# Structure of a Sulfated Capsular Polysaccharide from the Marine Bacterium *Cobetia marina* KMM 1449 and a Genomic Insight into Its Biosynthesis

**DOI:** 10.3390/md23010029

**Published:** 2025-01-08

**Authors:** Maxim S. Kokoulin, Yulia V. Savicheva, Alina P. Filshtein, Ludmila A. Romanenko, Marina P. Isaeva

**Affiliations:** G.B. Elyakov Pacific Institute of Bioorganic Chemistry, Far Eastern Branch, Russian Academy of Sciences, 159/2, Prospect 100 Let Vladivostoku, Vladivostok 690022, Russia; iu.savicheva0@yandex.ru (Y.V.S.); alishichka@mail.ru (A.P.F.); lro@piboc.dvo.ru (L.A.R.)

**Keywords:** marine bacteria, *Cobetia*, sulfated capsular polysaccharide, NMR, genome sequence, biosynthetic gene cluster

## Abstract

Some marine and extremophilic microorganisms are capable of synthesizing sulfated polysaccharides with a unique structure. A number of studies indicate significant biological properties of individual sulfated polysaccharides, such as antiproliferative activity, which makes them a promising area for further research. In this study, the capsular polysaccharide (CPS) was obtained from the bacterium *Cobetia marina* KMM 1449, isolated from a marine sediment sample collected along the shore of the Sea of Japan. The CPS was isolated by saline solution, purified by a series of chromatographic procedures, and studied by chemical methods along with 1D and 2D ^1^H and ^13^C NMR spectroscopy. The following new structure of the CPS from *C. marina* KMM 1449 was established and consisted of sulfated and simultaneously phosphorylated disaccharide repeating units: →4)-α-L-Rha*p*2S-(1→3)-β-D-Man*p*6PGro-(1→. To elucidate the genetic basis of the CPS biosynthesis, the whole genomic sequence of *C. marina* KMM 1449 was obtained. The CPS biosynthetic gene cluster (BGC) of about 70 genes composes four regions encoding nucleotide sugar biosynthesis (dTDP-Rha and GDP-Man), assembly (GTs genes), translocation (ABC transporter genes), sulfation (PAPS biosynthesis and sulfotransferase genes) and lipid carrier biosynthesis (*wcb* operon). Comparative analysis of the CPS BGCs from available *Cobetia* genomes showed the presence of KMM 1449-like CPS BGC among strains of all three *Cobetia* species. The study of new natural sulfated polysaccharides, as well as the elucidation of the pathways of their biosynthesis, provides the basis for the development of potential anticancer drugs.

## 1. Introduction

Oceans and seas cover approximately 70% of the Earth’s surface, providing a vital environment for a huge diversity of marine organisms and microbes. Microbial and genomic studies reveal a wide variety of marine bacterial groups throughout the water column and sediments, constituting up to 50% of the biosphere’s biomass [1,2]. In recent years, marine environments have emerged as significant sources of microorganisms that produce exopolysaccharides (EPSs). EPSs are water-soluble glycans produced by microorganisms and are found to accumulate outside their cells. They can be classified into two categories: capsular polysaccharides (CPSs), which are attached to the bacterial cell wall, and medium-released polysaccharides, which are released into the surrounding environment [3]. The EPSs derived from different bacterial sources exhibit a variety of structural combinations, which contributes to their unique properties [4,5].

In recent years, marine bacterial EPSs have attracted considerable interest due to their diverse structural and functional properties [3]. These substances possess various biological activities that make them valuable for a range of industrial applications, including pharmaceuticals, food products, cosmetics, and more [6]. Additionally, their potential for discovering novel molecules to combat bacteria, fungi, and viruses has further fueled research in this field [7].

Gram-negative bacteria of the genus *Cobetia* are obligate marine microorganisms that are widely distributed in aquatic environments [8]. Currently, the genus *Cobetia* comprises only three validly described species [8]. In the studied strains of *Cobetia*, the carbohydrate-containing polymers are acidic due to the presence of uronic acids, specifically 3-deoxy-D-manno-oct-2-ulosonic acid (Kdo), as well as non-carbohydrate components such as pyruvic acid and sulfates [9,10,11,12]. Sulfated polysaccharides are recognized for their significant antitumor effects, demonstrating antiproliferative, antimetastatic, pro-apoptotic, and anti-angiogenic properties. This opens up new possibilities for the biotechnological application of these halophilic microorganisms [13,14,15].

This study aimed to determine the structure of the simultaneously sulfated and phosphorylated CPS of *C. marina* KMM 1449, isolated from a marine sediment sample collected along the shore of the Sea of Japan. Additionally, this research sought to predict its biosynthesis using whole-genome sequencing combined with comprehensive bioinformatics analysis.

## 2. Results

### 2.1. Isolation and Structural Analyses of the CPS

The dried bacterial cells of strain KMM 1449 were extracted with saline solution, followed by enzymatic treatment and ultracentrifugation. The resulting CPS was purified by a series of chromatographic procedures, namely hydrophobic interaction chromatography, anion-exchange chromatography, and size-exclusion chromatography. The toluidine blue-stained SDS-PAGE of the purified CPS exhibited a broad smear in the upper part of the gel, which is typical for sulfated polysaccharides. The presence of sulfate groups was confirmed through FTIR spectroscopy, where a characteristic band was observed at 1253 cm^−1^, along with data obtained from the turbidimetric analysis.

A sugar analysis of the CPS by GC-MS of acetylated alditols showed the presence of rhamnose (Rha), mannose (Man), and a minor amount of glucose (Glc). A ratio of Rha and Man residues of ~1.7:1 (detector response) does not accurately reflect the sugar content, as further studies showed that the CPS is phosphorylated and the monosaccharides are not fully released from their respective phosphates. The D-configuration of Man and L-configuration of Rha were both determined by GC-MS of acetylated (S)-2-octylglycosides using the corresponding authentic samples [16].

The ^1^H and ^13^C NMR spectra of the CPS contained signals of different intensities, indicating structural heterogeneity. In detail, the ^13^C NMR spectrum of the CPS (Figure 1B) showed, inter alia, signals of two major anomeric carbons at δ_C_ 101.8 and 95.3, one methyl group of 6-deoxy sugar at δ_C_ 18.1, two hydroxymethyl groups of hexoses (data of the DEPT-135 experiment) at δ_C_ 65.9 and 62.3 (minor), and three characteristic signals of glycerol (Gro) residue at δ_C_ 67.7, 63.3 (hydroxymethyl groups, data of the DEPT-135 experiment), and 71.9. The spectra also contained minor signals, of which three at δ_C_ 25.7, 100.7, and 174.0 belonged to an acetal-linked pyruvic acid residue (Pyr), and the six at δ_C_ 106.1, 75.4, 73.8, 77.3, 66.6, and 65.9 were consistent with published data for 4,6-O-[(S)-1-carboxyethylidene]-β-D-glucose [17,18].

Accordingly, the ^1^H NMR spectrum of the CPS (Figure 1A) revealed, inter alia, signals for two major protons in the anomeric region at δ_H_ 4.61 and 4.28 (one more anomeric proton at δ_H_ 4.67 overlapped with HDO from the solvent), as well as one methyl group of 6-deoxy sugar at δ_H_ 1.34, and one minor methyl group of Pyr at δ_H_ 1.57. The ^31^P-NMR spectra of the CPS showed a signal for a phosphate group at δ_P_ 1.09. One reason for the CPS’s structural irregularity was likely due to non-stoichiometric phosphorylation, and therefore, the CPS was dephosphorylated for further analysis.

The ^13^C-NMR spectrum of the dephosphorylated CPS (dPCPS, Figure 2B) compared to the spectrum of native CPS contained, inter alia, signals for three main anomeric carbons with different intensities at δ_C_ 101.8 (**B-1**), 97.5 (**A′-1**), and 95.3 (**A-1**), one methyl group of 6-deoxy sugar at δ_C_ 18.1 (**A-6**), and one hydroxymethyl group of hexose (data of the DEPT-135 experiment) at δ_C_ 62.3 (**B-6**). The minor signals of 4,6-O-[(S)-1-carboxyethylidene]-β-D-glucose (**G**) were also observed. The ^1^H NMR spectrum of the dPCPS (Figure 2A) displayed signals for three protons in the anomeric region at δ_H_ 5.27 (**A-1**), 4.97 (**A′-1**), and 4.88 (**B-1**), one methyl group of 6-deoxy sugar at δ_H_ 1.33 (**A-6**), and one minor methyl group of Pyr at δ_H_ 1.57.

The ^1^H and ^13^C NMR spectra of the dPCPS were assigned by ^1^H, ^1^H-COSY, ^1^H, ^1^H-TOCSY, and ^1^H,^13^C-HSQC (Figure 3) experiments, and the chemical shifts are reported in Table 1. The signals within each spin system were assigned using the ^1^H, ^1^H-COSY, ^1^H, and ^1^H-TOCSY spectra, and the identities of L-Rha (two spin systems **A** and **A′**), D-Man (**B**), and D-Glc residues (**G**) were established by the characteristic coupling patterns. The α-linkage of L-Rha (**A** and **A′**) and the β-linkages of D-Man (**B**) and of D-Glc (**G**) residues were established by the ^1^J_C1–H1_ coupling constant values (172, 161, and 164 Hz, respectively) determined from the gated-decoupling experiment.

Linkage and sequence analyses of the dPCPS were performed using the ^1^H,^1^H ROESY, and ^1^H,^13^C-HMBC experiments (Figure 4). The ^1^H,^1^H ROESY spectrum revealed the following cross-peaks: H-1 **A**, **A′**/H-3 **B** at δ_H_/δ_H_ 5.27, 4.97/3.71, and H-1 **B**/H-3 **A**, **A′** at δ_H_/δ_H_ 4.88/3.68 (Figure 4A). Accordingly, the ^1^H,^13^C HMBC spectrum displayed the following inter-residue cross-peaks: H-1 **A**, **A′**/H-3 **B** at δ_H_/δ_C_ 5.27, 4.97/78.8, H-1 **A′**/H-3 **B** at δ_H_/δ_C_ 4.97/78.8, and H-1 **B**/H-3 **A**, **A′** at δ_H_/δ_C_ 4.88/80.6 (Figure 4B). The downfield position of the signal for C-2 of the α-L-Rha (**A**) residue at δ_C_ 78.9, in the ^13^C NMR spectra of the dPCPS, as compared with their position of residue **A′**, indicated the location of the sulfate group; therefore, the presence of the **A′** spin system is due to the partial cleavage of the sulfate group under the acidic conditions of the dephosphorylation procedure. The positions of the acetal carbon and the carboxyl group of Pyr were followed by the H-3/C-2 and H-3/C-1 correlations in the ^1^H,^13^C HMBC spectrum at δ_H_/δ_C_ 1.57/100.7 and 1.57/174.0, respectively. The location of Pyr was demonstrated by the **G** H-6/Pyr C-2 correlation at δ_H_/δ_C_ 4.13/100.7 in the ^1^H,^13^C HMBC spectrum (Figure 4C). Moreover, the ^1^H,^13^C HMBC spectrum displayed a cross-peak between H-1 of residue **G** at δ_H_ 4.67 and carbon at δ_C_ 81.6. Beginning at the cross-peak observed at δ_H_/δ_C_ 4.13/81.6, we identified an additional spin system corresponding to the terminal α-L-Rha (tA) residue. We can only speculate that the residue G is linked to the non-reducing end of the polysaccharide chain. However, further detailed studies are needed to confirm this hypothesis.

Based on the data of the chemical shifts obtained for the dPCPS, the native CPS was studied by 2D NMR spectroscopy, as described above, and additional spin systems for the phosphorylated β-D-Man and Gro residues were detected (Table 1). The downfield position of the signal for the C-6 of β-D-Man residue at δ_C_ 65.9 in the ^13^C NMR spectra of the CPS indicated the site of phosphorylation.

Therefore, the repeating unit of the CPS had the following structure:→4)-α-L-Rha*p*2S-(1→3)-β-D-Man*p*6PGro-(1→

### 2.2. Phylogenetic and Genomic Analyses

The complete genome of strain KMM 1449 was de novo assembled into one chromosome with an estimated size of 4,184,796 bp and an average GC content of 62.5%. Based on the RAST (Rapid Annotation using Subsystem Technology) and NCBI PGAP (Prokaryotic Genome Annotation Pipeline) results, the chromosome contains about 3370 coding gene sequences (ORFs), 21 rRNA genes (seven of 16S-23S-5S operons), and 72 tRNAs. The KMM 1449 genome sequence has been submitted to the NCBI submission portal under accession number CP173426.1.

The genome of strain KMM 1449 codes seven copies of the 16S rRNA gene, showing 100% sequence similarity to each other and to that of *C. marina* DSM 4741^T^ (=JCM 21022^T^). Also, they shared 99.52% sequence similarity with *C. amphilecti* KMM 1561^T^ (NRIC 0815^T^) one. The phylogenomic tree of KMM 1449, including the *Cobetia* type strains, clearly demonstrated that KMM 1449 falls into the *C. marina* clade (Figure 5).

Moreover, the average nucleotide identity (ANI) value between strains KMM 1449 and *C. marina* JCM 21022^T^ was 98.44%, which exceeds the 95-96% ANI threshold value accepted for species delineation [20]. The digital DNA–DNA hybridization (dDDH) value (formula d4) between strain KMM 1449 and *C. marina* JCM 21022^T^ was 86.2%, which was above the 70% threshold adopted for species differentiation [21]. These results indicate that KMM 1449 belongs to *C. marina*. To clarify the phylogenetic relationships among the *Cobetia* strains containing the KMM 1449-like polysaccharide genetic cluster, five *Cobetia* genomes were retrieved from NCBI datasets and included in the phylogenomic analysis. Two other *Cobetia* genomes from strains KMM 3879 and KMM 3880 were additionally selected for genomic analysis (Figure 5) due to their previously established polysaccharide structures [10,12].

### 2.3. Predicted CPS Gene Cluster and Biosynthesis

The search for CPS-related genes was based on the established composition of the CPS disaccharide fragment (Table 1). In the first step, based on the antiSMASH results, a genomic region of 93,710 bp in length was identified as a potential CPS gene cluster due to the presence of genes encoding the synthesis of activated nucleotide sugars, dTDP-Rha and GDP-Man (Figure 6). The operon *rmlBDAC* for the dTDP-L-Rha biosynthesis was found adjacent to a gene encoding a fusion protein comprising mannose-1-phosphate guanylyltransferase and mannose-6-phosphate isomerase domains, which may be responsible for the GDP-D-Man biosynthesis.

According to Table 1, the CPS structure contains D-Glc residues; however, the gene encoding an enzyme (GalF, UTP—glucose-1-phosphate uridylyltransferase) for activation of glucose was located at a distance of 583,801 bp from the CPS gene cluster. A search for polysaccharide export genes revealed the presence of *kpsETMD* operon for the group III CPS ABC transporter [22,23]. In the opposite direction was the *kpsSC* operon, which is responsible for the Kdo priming of a lipid carrier. Between these operons, PAPS biosynthesis-related genes were revealed, and the *wcb* operon was identified downstream to *kpsSC*. In addition, the genes coding for assembly (GTs, glycosyltransferases) and sulfation (ST, sulfotransferase) of the polysaccharide were also found.

**Figure 6 marinedrugs-23-00029-f006:**
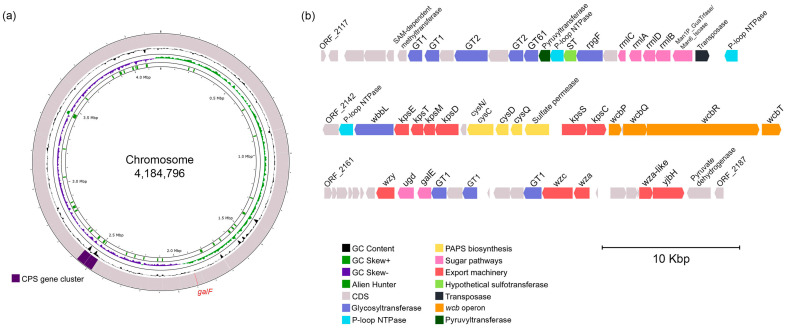
Chromosome location (**a**) and gene cluster organization (**b**) for the KMM 1449 CPS biosynthesis. Visualization was performed on the Proksee server [24]. The scale is shown in megabases (Mbp) for chromosomes and in kilobases (Kbp) for gene clusters. A gene for UDP-D-Glc biosynthesis is shown in red on chromosomes.

To elucidate the metabolic pathways for the synthesis of activated nucleotide sugars that participate in the formation of the CPS structural unit, the genome of strain KMM 1449 was investigated in detail (Figure 7). Based on the KEGG annotation [25,26], strain KMM 1449 is capable of utilizing maltose, sucrose, mannitol, fructose, and xylose since it has the relevant transporters and enzymes for these processes. The simplified scheme of the putative metabolism of the sugars indicates that the KMM 1449 genome also contains all the genes required for sugar conversion into nucleotide sugar precursors.

### 2.4. Comparative Analysis of CPS Gene Clusters Between Cobetia Strains

To date, there are only a few publications devoted to establishing the polysaccharide structures from *Cobetia* strains [9,10,11,12], but there is no publication dedicated to establishing their biosynthetic gene clusters (BGCs). Here, eight *Cobetia* genomes were taken for a comparative analysis of their CPS BGCs, focusing on the already elucidated organization of this cluster in the KMM 1449 genome. The main criteria for the genome selection were gene cluster integrity (1), the presence of the rhamnose operon (2) and *wcb* operon (3) near the ABC transporter encoding genes (4), as well as the PAPS biosynthesis genes (5). Two genomes of strains KMM 3879 and KMM 3880 were also taken into the comparative analysis since their polysaccharide structures were previously established [10,12]. The general characteristics of the genomes are summarized in Table 2.

The comparative analysis allowed us to determine common boundaries for all *Cobetia* CPS BGCs: SAM-dependent methyltransferase was at the left, and Pyruvate dehydrogenase was at the right (Figure 8). Conditionally, these gene clusters can be divided into five regions based on their gene composition and functions.

Region 1 composes genes responsible for the synthesis and activation of nucleotide sugars, the biosynthesis of a polysaccharide chain (GTs genes), as well as for its modification (ST, pyruvyltransferase, and several P-loop NTPases).

Region 2 contains genes that are specific for the coding of the ABC transporter of group III CPS. Typically, the genomic organization of the CPS III group clusters is represented by conserved regions 1 and 3 and variable region 2 [22,23]. Region 1 (in our case, an operon 1) is composed of *kpsDMTE* genes encoding KpsE, M, and D proteins for the formation of a channel across the inner and outer membranes, and KpsT is involved in ATP hydrolysis for substrate transport. Region 3 (=operon 3) is located in the opposite direction from region 1 on the leading chain and consists of *kpsSC* genes, encoding KpsSC, respectively, providing the synthesis of a 2-keto-3-deoxyoctonic acid (KDO) priming chain for the growing polysaccharide on a lipid carrier [27]. Region 2 (=operon 2) contains a bifunctional *cysN/cysC* (Sulfate adenylyltransferase, subunit 1 (EC 2.7.4.4)/adenylylsulfate kinase (EC 2.7.1.25)), *cysD* (Sulfate adenylyltransferase, subunit 2 (EC 2.7.4.4)), *cysQ* (3′-phosphoadenosine 5′-phosphate phosphatase (EC:3.1.3.7)), and a sulfate permease gene. This gene set encodes the proteins responsible for the biosynthesis of PAPS (3′-phosphoadenosine-5′-phosphosulfate), which is a universal sulfate donor. After a sulfate group is transferred from PAPS to the substrate by sulfotransferases, the remaining PAP is dephosphorylated to AMP by CysQ [28,29,30].

Region 3 represents a *wcb* operon consisting of *wcbPQRT* and encoding proteins for capsular polysaccharide lipid biosynthesis. This operon has been described for a pathogenic bacterium *Burkholderia pseudomallei* (the class *Betaproteobacteria*); however, little is known about the functions of these homologous proteins [31,32,33]. According to our results, the *wcb* operon was frequently revealed among members of the genus *Cobetia* (the class *Gammaproteobacteria*). In addition, it was reported that WcbQ, annotated with InterProScan as lipoteichoic acid (LTA) synthase, is able to transfer glycerol onto the polysaccharide chain using a membrane lipid phosphatidylglycerol as a GroP (glycerol phosphate) donor [34].

Region 4 was predicted to synthesize O-antigen due to the presence of genes encoding activated sugar biosynthesis, GTs, and wzy-dependent pathway. However, a Wzx-flippase was identified only in the *Cobetia* sp. D5 CPS BGS. Based on the NCBI BLAST results, the homologous oligosaccharide flippase originated from *Zeimonas* sp. (the class *Betaproteobacteria*) with 44.4% identity. If known O-antigenic Wzx-flippases are not found in the *Cobetia* genomes, then some unknown protein, designated here as hypothetical, can be used to translocate the O-antigen oligosaccharide chain into the periplasm.

According to the comparative analysis of KMM 1449 CPS-like BGCs of the *Cobetia* representatives, the highest nucleotide identity was observed in regions 2 and 3. This indicates conservative pathways used for the biosynthesis of a lipid carrier (probably fatty acids), translocation by the ABC transporter, and sulfation of polysaccharides. At the same time, a comparison of the organization of region 1, which is responsible for the synthesis of monosaccharides and the assembly of polysaccharides, allows us to assume a similar CPS structure in strains D5, N-80, ena-yuan-GCF_007786215.1, MM1IDA2H-1AD, and cqz5-12 with the structure of KMM 1449 CPS.

Remarkably, the positions of the ST genes adjacent to the genes of RgpF (GT) and RmlB, D, A, and C (dTDP-L-Rha-biosynthesis) also indicate that L-rhamnose will be sulfated at the second position in the CPS in all five *Cobetia* strains presented above.

It should be noted that in the KMM 3879 genome, the rhamnose operon was found in the CPS BGC (region 1) and not in the O-antigen BGC (region 4) (Figure 8). Moreover, the genes responsible for the biosynthesis of activated nucleotide glucose and galactose were absent in the CPS/ O-antigen BGC. In the KMM 3880 genome, the genes responsible for the synthesis of Kdo and GDP-D-Man were successfully identified, with the exception of *galF* (UDP-D-Glc), which was located outside of the CPS/ O-antigen BGC. Based on structural and genomic data, we can suggest that the established O-antigen structures [10,12] are actually the CPS structures.

Thus, it can be summarized that using only bioinformatic methods, it is not always possible to predict the exact structure of polysaccharides. In addition, the integration of bioinformatics, spectroscopy, and molecular biology approaches might clarify insight into the basic mechanisms and functions of glycosyltransferases and other proteins involved in polysaccharide assembly.

## 3. Materials and Methods

### 3.1. Isolation and Purification of CPS

Strain *C. marina* KMM 1449 was obtained from the Collection of Marine Microorganisms (KMM) at the G.B. Elyakov Pacific Institute of Bioorganic Chemistry, Far East Branch of the Russian Academy of Sciences (Vladivostok, Russia). The bacterium was cultivated for 48 h at ambient temperature in a medium (12 L) consisting of 5.0 g of bacto peptone, 1.0 g of yeast extract, 1.0 g of glucose, 0.2 g of K_2_HPO_4_, 0.05 g of MgSO_4_ and 750 mL of natural seawater/250 mL of distilled water at pH 7.8. Dry bacterial cells (10 g) were suspended in 100 mL of extraction buffer (0.22 M NaCl, 0.026 M MgCl_2_, 0.01 M KCl) and stirred continuously for 16 h at 4 °C. The cell pellet was collected by centrifugation (5000 rpm, 25 min, 4 °C), and the extraction procedure was repeated two more times (4 °C, 1 h). Supernatants were combined, dialyzed (MWCO 12,000 Da), and lyophilized to yield crude CPS (513 mg). The crude CPS (100 mg) was subjected to anion-exchange chromatography on a column (10 × 1.5 cm) using the Toyopearl DEAE-650M (Tosoh Bioscience, Tokyo, Japan) in a step-wise gradient of NaCl. The resulting main fraction (56.6 mg, 0.5 M NaCl) was dialyzed and freeze-dried. After that, the CPS was purified by hydrophobic interaction chromatography using a Butyl Sepharose 4FF column (10 × 1.5 cm, Sigma, St. Louis, MO, USA) with the method described in [35]. The CPS was eluted with buffer in a non-bound fraction. Finally, the CPS was desalted on the Toyopearl HW-40 column (120 × 1.5 cm, Tosoh Bioscience, Tokyo, Japan) and eluted with aq 0.3% AcOH, yielding the final CPS (42 mg). The elution was monitored with a differential refractometer (Knauer, Berlin, Germany).

### 3.2. Compositional Analysis of CPS Samples

Monosaccharides were analyzed as acetylated aldidols using the appropriate authentic samples as a reference. The absolute configurations of the sugar residues were determined through GC-MS of the acetylated (S)-2-octyl glycosides, as described in [16]. All the derivatives were analyzed using a Hewlett Packard 5890 chromatograph (Conquer Scientific, Poway, CA, USA) equipped with a Hewlett Packard 5973 mass spectrometer and a HP-5MS capillary column. Acetylated methyl glycosides were analyzed using the following temperature program: 150 °C for 3 min, 150 °C → 280 °C at 3 °C/min, and 280 °C for 10 min. For the acetylated (S)-2-octyl glycosides, the analysis was performed at 160 °C for 3 min, 160 °C → 290 °C at 3 °C/min, and 290 °C for 10 min.

### 3.3. Dephosphorylation of CPS

The CPS sample (20 mg) was dephosphorylated with aq 48% HF (0.2 mL, 4 °C, 16 h), HF was evaporated, and dPCPS (9 mg) was isolated in the Toyopearl HW-40 column (120 × 1.5 cm, Tosoh Bioscience, Tokyo, Japan), eluted with aq 0.3% AcOH and monitored as above.

### 3.4. NMR and FTIR Spectroscopy

The ^1^H and ^13^C NMR spectra of the CPS were recorded on a Bruker Avance-III (700.13 MHz for ^1^H and 176.04 MHz for ^13^C) spectrometer (Bruker, Karlsruhe, Germany) at 37 °C using acetone (δ_C_ 31.45, δ_H_ 2.225) as an internal standard. The ^1^H,^1^H-TOCSY, and ^1^H,^1^H-ROESY spectra were recorded with a 180 ms duration of MLEV-17 spin-locking and a 200 ms mixing time, respectively. ^1^H,^13^C-HMBC was optimized for an 8 Hz long-range constant. The FTIR spectrum of the CPS was registered in a KBr pellet on a Vector 22 Fourier-transform spectrophotometer (Bruker, Karlsruhe, Germany) with a 4 cm^−1^ resolution.

### 3.5. Whole-Genome Sequencing, Assembly, and Annotation

Genomic DNA was extracted from strain KMM 1449 using the NucleoSpin Tissue kit (Macherey–Nagel, Düren, Germany), and the nanopore library was prepared using the SQK-RAD002 kit (Oxford Nanopore Technologies, Oxford, UK) and sequenced on MinION (Oxford Nanopore Technologies, Oxford, UK). Genome assembly was performed using Flye, version 2.9 [36], with the default parameters. The genome completeness and contamination were estimated by CheckM version 1.1.3 [37]. Genome annotation was carried out using the NCBI Prokaryotic Genome Annotation Pipeline (PGAP) [38], Rapid Annotation using Subsystem Technology (RAST) [39], Prokka [40], eggNOG [41], InterPro [42], and UniProt [43]. The annotation of secondary metabolite biosynthetic gene clusters was conducted using antiSMASH server version 7.0 [44]. The circular genome of strain KMM 1449 and the CPS gene cluster structure were both visualized using the Proksee platform [24]. Comparison of the ANI and dDDH of the strains KMM 1449 and *C. marina* JCM 21022^T^ was performed with the online server EzBioCloud and Type (Strain) Genome Server (TYGS) (https://tygs.dsmz.de/, accessed on 30 November 2024) platform, respectively [45,46]. The phylogenomic tree of KMM 1449 with *Cobetia* strains was constructed using the TYGS platform [46]. A pairwise comparison between CPS gene clusters of the *Cobetia* strains was carried out using BLASTn (BLAST version 2.16.0+) run in EasyFig (version 2.2.5) [47]. The KAAS–KEGG Automatic Annotation Server was used to elucidate the sugar metabolic pathways [26].

## 4. Conclusions

Sulfated polysaccharides are found in various microorganisms, particularly in many marine Gram-negative bacteria. These charged biomolecules are thought to enhance the ability to bind with naturally occurring cations. The presence of charged molecules allows for the formation of a rigid network of cross-links, which provides greater resistance to physical stress and helps maintain the proper physiology and functionality of bacterial cells in marine environments. Research has shown that some sulfated polysaccharides can be cytotoxic and inhibit the growth of human cancer cells [48,49]. As a result, sulfated polysaccharides derived from marine bacteria represent a promising area of research in biomedicine [50].

In summary, we characterized the structure of the CPS from *C. marina* KMM 1449, isolated from a marine sediment sample. The polysaccharide was identified as a novel sulfated and phosphorylated bioglycan, and its disaccharide repeating unit possesses a unique structure not found among known bacterial polysaccharides (http://csdb.glycoscience.ru/bacterial accessed on 18 December 2024) [51]. The sequencing and chromosome assembly of the strain KMM 1449 genome was performed to elucidate the CPS BGC organization. The BGC of about 70 genes is composed of three regions encoding nucleotide sugar biosynthesis (dTDP-Rha and GDP-Man), assembly (GTs genes), translocation (ABC transporter genes), sulfation (PAPS biosynthesis and sulfotransferase genes), and lipid carrier biosynthesis (*wcb* operon). Comparative analysis of the CPS BGCs from available *Cobetia* genomes showed the presence of KMM 1449-like CPS BGC among strains of all three *Cobetia* species.

## Figures and Tables

**Figure 1 marinedrugs-23-00029-f001:**
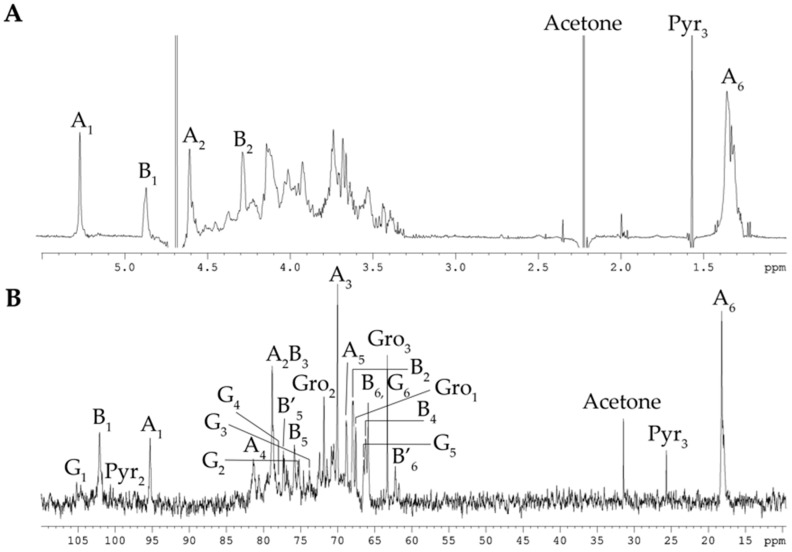
^1^H NMR spectrum (**A**) and ^13^C NMR spectrum (**B**) of the CPS from *C. marina* KMM 1449. Numerals refer to carbons and protons in sugar residues denoted by capital letters, as described in Table 1.

**Figure 5 marinedrugs-23-00029-f005:**
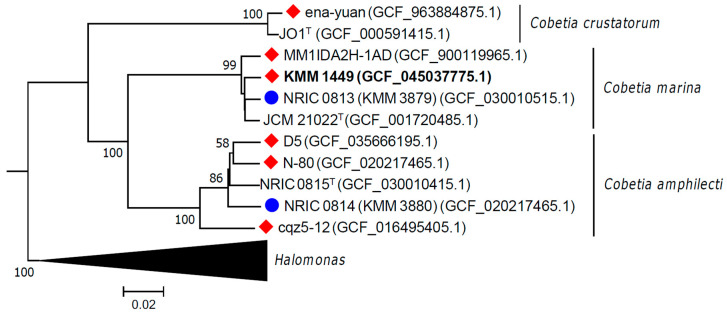
Genomic tree of *Cobetia* strains inferred with FastME 2.1.6.1 [19] based on GBDP distances (formula d5). GBDP pseudo-bootstrap support values are shown >50% from 100 replications. The average branch support was 83.5%. The tree was rooted at the midpoint. Strains with KMM 1449-like CPS gene cluster organization are marked with a red diamond, and strains with known polysaccharide structures are marked with a blue circle. KMM 1449 is marked in bold.

**Figure 7 marinedrugs-23-00029-f007:**
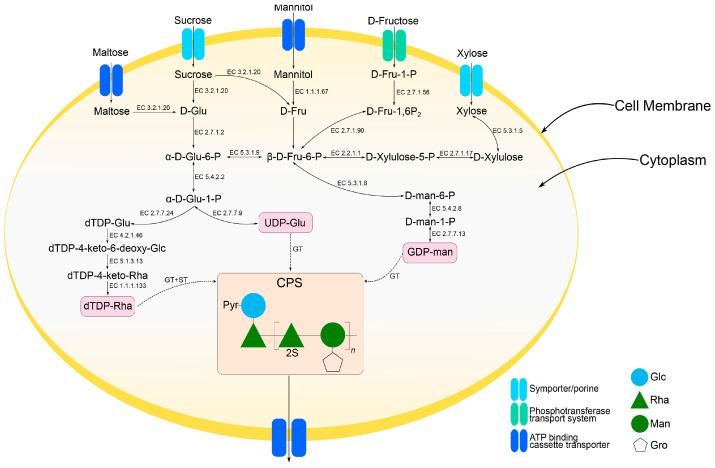
A scheme of metabolic pathways for the biosynthesis of activated nucleotide sugars for *Cobetia* KMM 1449 CPS obtained from genomic sequence data. The EC numbers identify the corresponding enzymes: EC 3.2.1.20 alpha-glucosidase; EC 2.7.1.2 glucokinase; EC 5.3.1.9 glucose-6-phosphate isomerase; EC 5.4.2.2 phosphoglucomutase; EC 2.7.7.9 UTP-glucose-1-phosphate uridylyltransferase; EC 2.7.7.24 glucose-1-phosphate thymidylyltransferase; EC 4.2.1.46 dTDP-glucose 4,6-dehydratase; EC 5.1.3.13 dTDP-4-dehydrorhamnose 3,5-epimerase; EC 1.1.1.133 dTDP-4-dehydrorhamnose reductase; EC 1.1.1.67 mannitol 2-dehydrogenase; EC 2.7.1.90 6-phosphofructokinase; EC 2.2.1.1 transketolase; EC 5.3.1.8 mannose-6-phosphate isomerase; EC 5.4.2.8 phosphomannomutase; EC 2.7.7.13 mannose-1-phosphate guanylyltransferase; EC 2.7.1.56 1-phosphofructokinase; EC 2.7.1.17 xylulokinase; EC 5.3.1.5 xylose isomerase; GT, glycosyltransferases; ST, sulfotransferase.

**Figure 8 marinedrugs-23-00029-f008:**
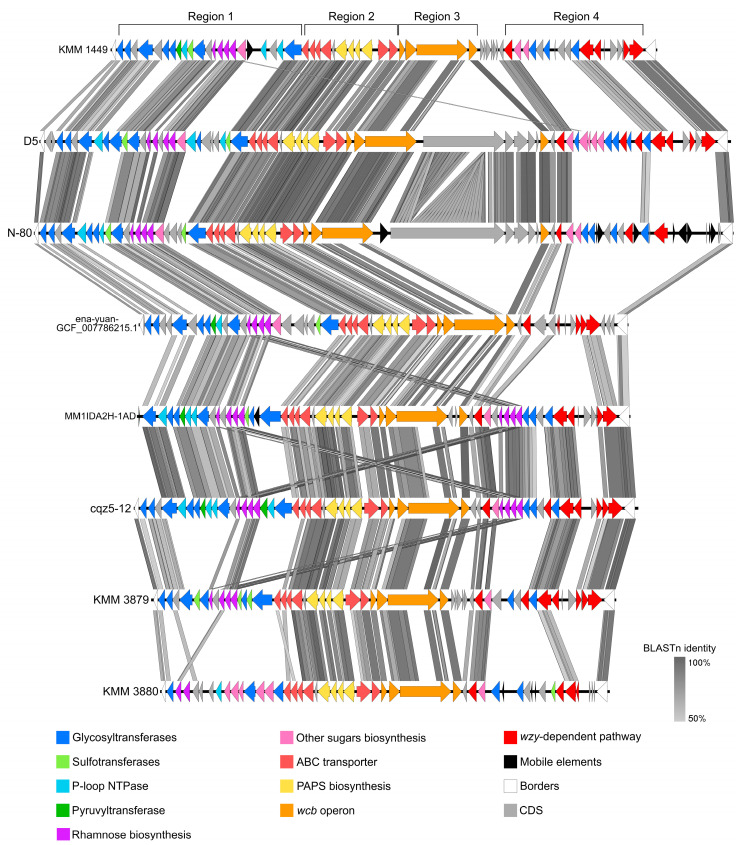
The comparisons of CPS gene loci between the *Cobetia* representatives. KMM 1449, *C. marina*; D5, *Cobetia* sp.; N-80, *C. amphilecti*; ena-yuan-GCF_007786215.1, *C. crustatorum*; MM1IDA2H-1AD, *C. marina*; cqz5-12, *Cobetia* sp.; and KMM 3879, *C. marina*; KMM 3880, *C. amphilecti*.

**Figure 2 marinedrugs-23-00029-f002:**
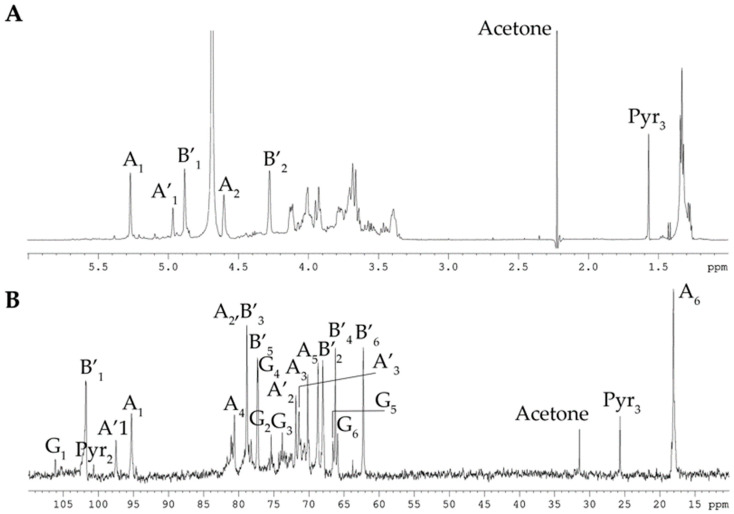
^1^H NMR spectrum (**A**) and ^13^C NMR spectrum (**B**) of the dPCPS from *C. marina* KMM 1449. Numerals refer to carbons and protons in sugar residues denoted by capital letters, as described in Table 1.

**Figure 3 marinedrugs-23-00029-f003:**
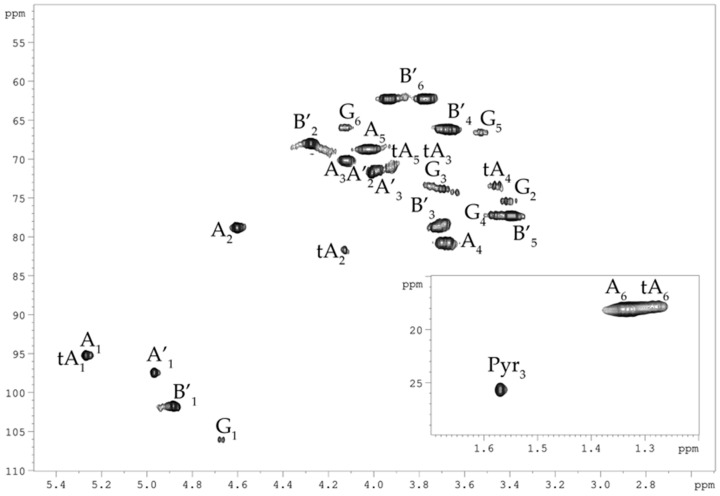
^1^H, ^13^C HSQC spectrum of the dPCPS from *C. marina* KMM 1449. Numerals refer to carbons and protons in sugar residues denoted by capital letters, as described in Table 1.

**Figure 4 marinedrugs-23-00029-f004:**
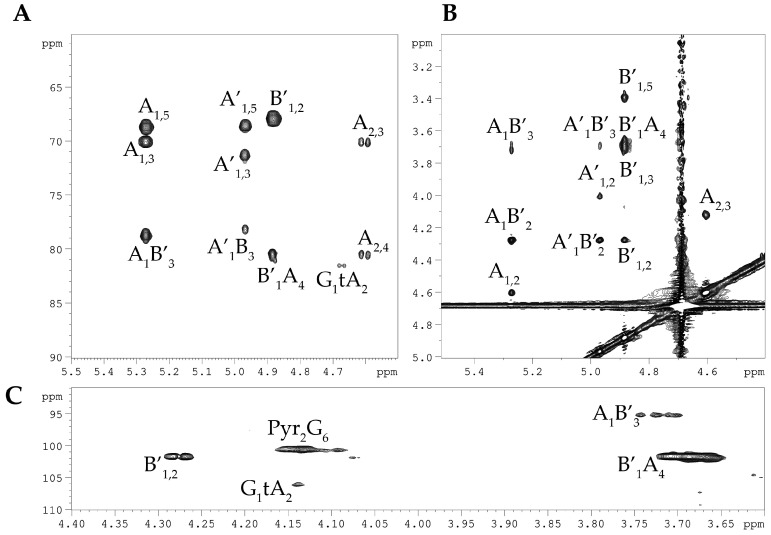
Fragments of the ^1^H, ^1^H ROESY spectrum (**A**) and ^1^H, ^13^C HMBC spectrum (**B**,**C**) of the dPCPS from *C. marina* KMM 1449. Numerals refer to carbons and protons in sugar residues denoted by capital letters, as described in Table 1.

**Table 1 marinedrugs-23-00029-t001:** ^1^H and ^13^C NMR data for the CPS and dPCPS from *C. marina* KMM 1449, *δ*, ppm.

Sugar Residue	H-1	H-2	H-3	H-4	H-5	H-6a,b
	C-1	C-2	C-3	C-4	C-5	C-6
→4)-α-L-Rha*p*2S-(1→	5.27	4.61	4.13	3.68	4.02	1.34
**A**	95.3	78.9	70.2	80.6	68.7	18.1
→4)-α-L-Rha*p*-(1→	4.97	4.01	3.98	3.68	4.02	1.34
**A′**	97.5	71.8	71.4	80.6	68.7	18.1
α-L-Rha*p*-(1→	5.29	4.13	3.93	3.46	3.91	1.28
**tA′**	95.3	81.6	71.2	73.2	71.2	18.1
→3)-β-D-Man*p*6PGro-(1→	4.88	4.28	3.71	3.67	3.53	4.14, 4.09
**B**	101.8	68.0	78.8	66.2	75.9	65.9
Gro-(1→	3.97, 3.97	3.92	3.69, 3.63			
	67.7	71.9	63.3			
→3)-β-D-Man*p*-(1→	4.88	4.28	3.71	3.67	3.39	3.94, 3.77
**B′**	101.8	68.0	78.8	66.2	77.3	62.3
β-D-Glc*p*4,6(*S*-Pyr)-(1→	4.67	3.42	3.69	3.45	3.53	4.13, 3.69
**G**	106.1	75.4	73.8	77.3	66.6	65.9
*S*-Pyr			1.57			
	174.0	100.7	25.7			

**Table 2 marinedrugs-23-00029-t002:** The accession numbers and general features of eight *Cobetia* genomes used in this study.

Strain	Accession ID	Genome Size (bp)	Number of Contigs	G+C (mol%)	CPS/O-Antigen Length (bp)	Number of CDSs in Cluster
*Cobetia marina* KMM 1449	GCF_045037775.1	4,184,796	1	62.5	87,222	65
*Cobetia* sp. D5	GCF_035666195.1	4,233,985	1	62.5	110,141	69
*Cobetia amphilecti* N-80	GCF_020217465.1	4,160,095	1	62.5	111,868	68
*Cobetia crustatorum* ena-yuan-GCF_007786215.1	GCF_963884875.1	4,215,468	163	57.5	78,218	56
*Cobetia marina* MM1IDA2H-1AD	GCF_900119965.1	4,155,178	105	62	82,584	57
*Cobetia* sp. cqz5-12	GCF_016495405.1	4,209,007	1	62.5	78,267	55
*Cobetia marina* NRIC 0814 (formerly *C. pacifica* KMM 3879^T^)	GCF_030010515.1	4,066,371	42	62.5	73,878	53
*Cobetia amphilecti* NRIC 0813 (formerly *C. litoralis* KMM 3880^T^)	GCF_029846315.1	4,621,254	51	62.5	71,482	48

## Data Availability

The GenBank accession number for chromosome strain KMM 1449 is CP173426.1. The GenBank/RefSeq assembly accessions are GCA_045037775.1 and GCF_045037775.1, respectively.

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
