# Peer review of "Structure of a Sulfated Capsular Polysaccharide from the Marine Bacterium Cobetia marina KMM 1449 and a Genomic Insight into Its Biosynthesis"

_marinedrugs, 2025, doi:10.3390/md23010029_

Round 1

Reviewer 1 Report

Comments and Suggestions for Authors

In this study, a novel sulfated capsular polysaccharide with sulfated and phosphorylated bioglycan was identified. This disaccharide repeating unit possesses a unique structure not found among known bacterial polysaccharides, which is a gratifying finding. The biosynthetic pathway was also proposed according to genomic analyses and it was presented among strains of all three Cobetia species.

I have some questions:

1, The importance to study the structure of sulfated capsular polysaccharides and its biosynthetic pathway should be pointed out in the absatract.

2, The name of genus “Cobetia” should be italic. Please check whole paper.

3, Line 46, authors say that genus Cobetia are widely distributed in aquatic environments. Attached related references or provide related evidences.

4, Line 153, the digital DNA–DNA hybridization (dDDH) values is also used to discriminate bacterial species. Please calculate.

5, Line 285, please confirm that the abbreviation of Collection of Marine Microorganism.

6, Not all sulfated polysaccharides possess medicinal activity. Although the composition and structure of the CPS from KMM 1449 are novel, there are other polysaccharides with similar structures and compositions. Have there been any reports on the medicinal value of these similar polysaccharides? Or have there been any preliminary experiments to explore the medicinal potential of the CPS from KMM 1449?

Author Response

Thank you very much for taking the time to review our manuscript.

Comment 1: The importance to study the structure of sulfated capsular polysaccharides and its biosynthetic pathway should be pointed out in the abstract.

Response 1: Thank you for your comment. We've added a few sentences to highlight this (Lines 11-14, 27, 28).

Comment 2: The name of genus “Cobetia” should be italic. Please check whole paper.

Response 2: Agree. The genus name has been checked and corrected to italics throughout the manuscript.

Comment 3: Line 46, authors say that genus Cobetia are widely distributed in aquatic environments. Attached related references or provide related evidences.

Response 3: Agree. The reference 8 below supports this claim. We have duplicated this reference, Line 53.

Comment 4: Line 153, the digital DNA–DNA hybridization (dDDH) values is also used to discriminate bacterial species. Please calculate.

Response 4: Thank you for your comment. We have added the dDDH value calculated between the KMM 1449 and JCM 21022T genomes using d4 formula, Lines 193-195.

Comment 5: Line 285, please confirm that the abbreviation of Collection of Marine Microorganism.

Response 5: The Collection of Marine Microorganisms is a member of the WDCM (official acronym KMM and registration number 644).

Comment 6: Not all sulfated polysaccharides possess medicinal activity. Although the composition and structure of the CPS from KMM 1449 are novel, there are other polysaccharides with similar structures and compositions. Have there been any reports on the medicinal value of these similar polysaccharides? Or have there been any preliminary experiments to explore the medicinal potential of the CPS from KMM 1449?

Response 6: Unfortunately, to the best of our knowledge, there are no data on the biological activity of similar polysaccharide structures. In the future, we plan to study a number of biological properties of these substances.

Reviewer 2 Report

Comments and Suggestions for Authors

Review for Structure of a sulfated capsular polysaccharide from the marine bacterium Cobetia marina KMM 1449 and a genomic insight into its biosynthesis by Maxim S. Kokoulin ,#, Yulia V. Savicheva#, Alina P. Filshtein, Ludmila A. Romanenko and Marina P. Isaeva  strong experience from this team with recent papers such as

 Capsular polysaccharide from the marine bacterium Cobetia marina induces apoptosis via both caspase-dependent and mitochondria-mediated pathways in HL-60 cells

Kokoulin, M.S., Kuzmich, A.S., Filshtein, A.P., Prassolov, V.S., Romanenko, L.A. Carbohydrate Polymers, 2025, 347, 122791

 __________________________

Proposal of Cobetia marina gen. nov., comb. nov., within the family Halomonadaceae, to include the species Halomonas marina

Arahal, D.R., Castillo, A.M., Ludwig, W., Schleifer, K.H., Ventosa, A. Systematic and Applied Microbiology, 2002, 25(2), pp. 207–211 Cobetia, new genus in 2002

previous works about sulfated capsular polysaccharide under the former genus name?

________________________________

The capsular polysaccharide (CPS) was obtained from a bacterium Cobetia marina KMM 1449, isolated from a marine sediment sample collected along the shore of the Sea of Japan this bacteria would be easy to produce in large scale fermenters?

__________________

so many bacterial sulfated capsular polysaccharides available, why choosing yours?

___________________________

Comparative analysis of the CPS BGCs from available Cobetia genomes showed the presence of KMM 1449-like CPS BGC among strains of all three Cobetia species, what about KMM 1449-like CPS BGC in close genera? in distant microorganisms?

___________________

In recent years, marine bacterial EPSs have attracted considerable interest due to their diverse structural and functional properties is there already some true PHARM applications? trademark, companies?

_____________________________

this is a very nice work, congrats to the authors 

Author Response

Thank you very much for taking the time to review our manuscript.

Comment 1: Proposal of Cobetia marina gen. nov., comb. nov., within the family Halomonadaceae, to include the species Halomonas marina Arahal, D.R., Castillo, A.M., Ludwig, W., Schleifer, K.H., Ventosa, A. Systematic and Applied Microbiology, 2002, 25(2), pp. 207–211 Cobetia, new genus in 2002.

previous works about sulfated capsular polysaccharide under the former genus name?

Response 1: We hope we have understood your comment correctly. No published data on the structures of sulfated polysaccharides from Halomonas marina (= Cobetia marina DSMZ 4741T) have been found. Among other Halomonas, the best known is mauran, a sulfated exopolysaccharide produced by Halomonas maura, for which neither the structure nor the gene cluster is yet known.

Comment 2: The capsular polysaccharide (CPS) was obtained from a bacterium Cobetia marina KMM

1449, isolated from a marine sediment sample collected along the shore of the Sea of Japan

this bacteria would be easy to produce in large scale fermenters?

Response 2: Experiments on the synthesis of polysaccharide in large quantities were not included in this study.

Comment 3: so many bacterial sulfated capsular polysaccharides available, why choosing yours?

Response 3: When studying KMM strains for the ability to produce sulfated polysaccharides, the presence of a sulfated polymer was revealed in strain KMM 1449.

Comment 4: Comparative analysis of the CPS BGCs from available Cobetia genomes showed the presence of KMM 1449-like CPS BGC among strains of all three Cobetia species, what about KMM 1449-like CPS BGC in close genera? in distant microorganisms?

Response 4: Homologous genes from the KMM 1449-like CPS BGC were mainly found among the genera Halomonas and Salinicola, but the entire KMM 1449-like cluster has not been revealed among published genomes.

Comment 5: In recent years, marine bacterial EPSs have attracted considerable interest due to their diverse structural and functional properties is there already some true PHARM applications? trademark, companies?

Response 5: So far, only two sulfated cyanobacterial EPSs are used as a moisturizer for skin wound healing (cyanoflan) and as a traditional food in Japan for the treatment of gastroenteritis and allergies (sacran).

Pandey S., Kannaujiya V. K. Bacterial extracellular biopolymers: Eco-diversification, biosynthesis, technological development and commercial applications. International Journal of Biological Macromolecules. 2024, 135261.
